# Setbacks to IoT Implementation in the Function of FMCG Supply Chain Sustainability during COVID-19 Pandemic

**Jelena Končar, Aleksandar Grubor, Radenko Marić \*, Sonja Vučenović and Goran Vukmirović**

Faculty of Economics in Subotica, University of Novi Sad; 24000 Subotica, Serbia; koncarj@ef.uns.ac.rs (J.K.); agrubor@ef.uns.ac.rs (A.G.); sonjavucenovic1@ef.uns.ac.rs (S.V.); vgoran@ef.uns.ac.rs (G.V.)

\* Correspondence: radenko.maric@ef.uns.ac.rs; Tel.: +381-62-278-559

**Abstract:** One of the basic measures of the World Health Organization (WHO) in the fight against the COVID-19 pandemic is a lockdown policy with reduced contacts and physical distance. This presents a challenge, especially for fast-moving-consumer-goods (FMCG) supply chains, which are characterized by a large number of physical contacts between employees in production, physical distribution, wholesale, and retail. One of the ways to comply with the prescribed measures with the smooth functioning of the supply chain is the complete digitalization and automation of all business activities and operations based on the application of the Internet of Things (IoT). In this regard, this paper aims to analyze the setbacks to the digitalization of business processes and the sustainability of the FMCG supply chain based on the implementation of IoT. The research has been conducted among the participants in the standardization chain in the sectors of production, physical distribution, wholesale, and retail of FMCG in the Western Balkans region during the COVID-19 pandemic. The results showed significant differences between business sectors in terms of the intensity of setbacks to successful IoT implementation. Based on the obtained results, a set of measures and incentives was proposed that the competent institutions and the management of the FMCG supply chain should apply to encourage the digitalization process. Suggestions for future research are given in the paper.

**Keywords:** FMCG supply chain; physical distribution; Internet of Things; COVID-19 pandemic; sustainability

---

## 1. Introduction

The global pandemic caused by the SARS-Cov-2 virus, named COVID-19 [1,2] was declared on 11 March 2020, by the World Health Organization (WHO). A pandemic is a geographical determinant that marks the spread of a contagious disease globally. In order to have pandemic potential, the disease must be widespread with great infectious potential, without established therapy, vaccine, or drug [3,4]. According to the WHO, the average daily growth rate of those infected in the period from the declaration of the pandemic to 24 July 2020, ranged from a minimum of 3% to a maximum of 17.15% (22 July). As a result of such a continuous growth trend, in 134 days from the beginning of the pandemic, the number of infected people increased from 126,215 to as many as 15,955,925 (24 July), while the total number of registered cases of coronavirus was recorded in 213 countries [5].

Since there is neither cure, vaccine, nor confirmed therapy for the coronavirus, the countries, in accordance with the WHO recommendations, have opted for various forms of social distancing, reduced contacts, and complete isolation. As a result of the measures taken, the trend of the exponential growth of the infection has been partially slowed down, however, such measures have caused a significant decrease in business activities in certain sectors, such as tourism, hotels, catering industry,

and retail [6]. Recent studies [7,8] point out that as a result of anti-pandemic measures, long-term market shocks, sudden changes in behavior patterns, and significant fluctuations in demand, have caused problems in the functioning of supply chains, especially the consumer goods chains (FMCG). There is a discrepancy between the supply and demand of basic foodstuffs (e.g., oil, flour, sugar, salt, milk, meat). There are delays and errors in deliveries, the unsustainability of production, disturbed storage conditions, etc. The efficiency, transparency, and sustainability of the FMCG supply chain are being questioned, because the lockdown policy as a basic measure against the COVID-19 pandemic, which was chosen by almost all European countries, affects the movement of people, business operations and leads to unnecessary (panic) stocking [9]. The authors agree that supply chains should be made flexible and sustainable, their resilience to sudden crises and market shocks should be increased, and we should make them less dependent on the human factor, through full digitalization and automation of business operations [10,11]. One of the ways to gradually digitalize the FMCG supply chain is the implementation of the Internet of Things (IoT) platform [12].

In academic research, there is an increasing number of papers dealing with the digitalization and introduction of IoT in production and service processes with the aim of encouraging sustainable development, eliminating errors made by the human factor, rationalizing energy use, optimizing cost, and improving business efficiency [13,14]. Digitalization is a trend that is imposed in modern conditions in all spheres of human activity. The COVID-19 pandemic is just the catalyst that hastened this trend and pointed to the need for digitalization. Research has shown that the implementation of digital strategies based on the IoT platform in supply chains gives concrete results. Research company for consulting in the Asian market—Tata Consultancy Services, surveyed 3764 supply chain managers for a 2016 report and found that 79% of them already use IoT to track final consumers, products, physical distribution, and outlets. The surveyed trading companies state that in retail facilities, which are included in their IoT initiatives, the average increase in sales volume in 2015 was 16%. In supply chains, which Tata Consultancy Services has identified as examples of good practice, the average increase in revenue has reached as much as 64% [15]. Forecasts show that by the end of 2020, between 18 and 50 billion products and service activities will be connected in the global market. This means that the IoT could become a market worth between $300 billion and $1.7 trillion [16,17].

The concept of IoT is based on the installation of a virtual platform that, through Radio Frequency Identification - RFID tags, bar codes, wireless sensors (WS), and smart devices, combines all the data on production and service operations within the supply chain [16]. All business processes and appliances connected to the IoT can be remotely controlled to achieve the desired functionality and sustainability of the supply chain [18]. In this way, the need for physical contact gets reduced, the system itself detects shortages, increased demand, stock position, and sends signals to the downward members of the supply chain promptly. This type of networking is especially emphasized in FMCG due to large daily fluctuations on the side of supply and demand. [19].

In recent years, in support of the IoT, a significant number of academic studies have indicated the application of several models for data and business process optimization. These are models that use different dimensions of data collected in real-time to predict future trends and business processes. These are models such as (1) multi-dimensional classification (MDC) [20,21], which observes each product from several dimensions. For example, the time dimension refers to the collection of product placement data depending on different weather dimensions or classes (morning, noon, evening, rain, sun, snow, fog, etc.) or, let's say, the language dimension which collects data about the influence of language labels (English, Chinese, Serbian, etc.) on product flow and so on. (2) Gaussian process models (GPM) [22], based on a large number of measurements, use real data to determine the relationship between tasks. For example, delay in physical distribution-state of the product range in retail facilities-consumer satisfaction, or maintaining optimal storage temperature-preservation of product use value-consumer satisfaction, etc. (3) Artificial neural network models (ANN) [23] based on interdependence between sets (nodes) of different data, predicts future trends, etc. The application of

these models complements the IoT, providing safer forecasting, more efficient placement mechanisms, required quantities in target market segments, and more transparent supply chains.

The COVID-19 pandemic will significantly reshape FMCG supply chains. The impact of the lockdown policy on the movement of people and business operations is so vast that some forecasts predict a decline in the trade sector by as much as 27% in the second quarter of 2020 [9]. The authors agree that IoT implementation would partially mitigate negative trends, make supply chains more flexible to current market developments, and better prepare them for the post-COVID-19 pandemic period [16,24].

The paper consists of five parts. After the introduction, the theoretical background points out the importance of IoT-based digitalization during the COVID-19 pandemic. The advantages of the digitalization process and the largest setbacks hindering its successful implementation are defined. The methodology section goes on to explain the aim of the research, research hypotheses, sample structure, research procedures, and applied statistical methods. The research results chapter summarizes the most important research results and testing of the set hypotheses. The discussion chapter compares the results of related studies and proposes a set of measures and recommendations for improving the digitalization process. The conclusion briefly summarizes the most important results of the study, points out the observed shortcomings, and gives suggestions for future research.

## 2. Theoretical Background

### 2.1. Subject, Goals, and Contributions of the Paper

In the Western Balkans (WB), the implementation of the IoT platform is at the very beginning due to the still outdated legal framework, untrained staff, lack of financial resources, lack of modern technology, consumer, and market distrust in this type of business [17]. The COVID-19 pandemic is a real test for FMCG supply chains of WB countries to digitalize their business processes, turn to modern technologies, and thus strengthen their competitive position in the global market. With this regard, the subject of this paper is the analysis of the importance of the implementation of the IoT concept for the sustainability of the FMCG supply chain during and after the COVID-19 pandemic.

The paper goals are (1) defining setbacks to the digitalization of business processes and the sustainability of the FMCG supply chain based on the implementation of the Internet of Things in the WB region. Then, (2) filling the research gap in WB countries, as only a few academic studies examine this issue. Finally, (3) unlike previous studies that dealt with individual obstacles to the implementation of IoT in supply chains, this paper categorizes all setbacks in one place and their real impact on delaying the implementation of IoT.

The need for the research stems from the fact that this is the market where the service sector dominates. Out of the total number of registered legal entities, 35% to 40% of registered companies are in the FMCG supply chain segment. As a percentage, the highest number of employees is in the FMCG placement sector (around 10%), with the share of FMCG retail revenues in the total GDP of the WB of 11%. Moreover, the WB region is interesting for research because it is a market of over 20 million consumers, characterized by huge economic and demographic differences across regions, making it suitable for comparison. As the production and marketing of FMCG are one of the main drivers of economic development in the Western Balkans, there is a need to start an in-depth analysis of the digitalization process in this sector during the COVID-19 pandemic.

The contributions of the paper include the following. (1) Defining a set of measures and incentives that the competent institutions and the management of the FMCG supply chain should apply to encourage the digitalization process; (2) setting a reliable research basis for future scientific research and academic studies related to the digitalization processes and implementation of IoT in supply chains; (3) proposing and encouraging innovative solutions to ensure greater flexibility of the FMCG supply chain during sudden market disturbances, such as the COVID-19 pandemic.

*2.2. Theoretical Basis of Formation and Structuring of the IoT implementation in the FMCG Supply Chain*

The COVID-19 pandemic has caused unforeseeable global consequences for the market, consumers, and social life. The lockdown policy, as the only measure in the fight against the pandemic, limited the movement of people, reduced the scope of business activities, and changed the patterns of behavior of consumers who turned to the accumulation of stocks in a panic. According to the third scenario of ending the pandemic, which is expected in the last quarter of 2020, a decline in global GDP of 10.7% is forecasted, namely, USA −8.5%, Germany −11.9%, China −5.8%, Japan −9.7%, France −10.6% [25]. Under these circumstances, the sustainability of FMCG supply chains is at stake.

The authors [9,26] point to some dangers facing supply chains. Panic among consumers and businesses has distorted common consumption patterns and created market anomalies. As the data for March and April show, the food and beverage sales increased by 20% in the UK [27] and by 25% in the USA [28]. The large-scale shortages of basic foodstuffs, medical equipment, medicines, and disinfectants occurred. The supply chain is not flexible enough and is not able to respond to sudden shocks that appear on the market. Retailers record increasing delays, errors, and costs in deliveries. As a result of the need for safe transport, it is necessary to meet several safety procedures and approvals that require much time, which significantly increases delays and costs for suppliers. At the same time, an increasing number of employees and companies turned to work from home, while consumers are increasingly opting for online ordering and electronic purchase of products. Consequently, Walmart e-retail grew by 74% in April 2020, while e-retail in the US specialty stores (food, beverages, drugs, etc.) grew at a rate of 141% in each month of the first quarter of 2020, while sales in traditional outlets fell by 1 to 5% [29]. As a result, retailers, as the last link in the FMCG supply chain, are closing traditional outlets, laying off workers, and massively accessing product returns. A major problem indicated by some studies [8] is the shortage of labor in the primary agricultural sectors for food production and processing, which rely on seasonal labor, due to worker illness, self-isolation, or restriction of movement. In addition, input delivery is limited for many companies, especially the supply of raw materials from countries that are hotspots of the COVID-19 pandemic. Many supply chain actors, primarily SMEs who do not have the versatility needed to maintain an acceptable level of supply, are unable to contribute to the sustainability of the chain in times of shock. They withdraw and leave the chains, challenging the viability of all other participants.

Recent research undoubtedly emphasizes the need to build a more flexible FMCG supply chain that would respond promptly to sudden market shocks [7]. It is necessary to develop a sustainable FMCG supply chain that will be more transparent and less dependent on the human factor. Kilpatrick [30] believes that it is necessary to transform traditional supply chains into digital supply networks (DSNs) with free flow of information and visibility from producers to end consumers. Digitalized chains can be implemented through advanced technologies, primarily IoT centralized platforms that integrate, process, and store all data on production and service processes, as well as the products, services, and people within the FMCG supply chain [7,12,31].

Singh et al. [31] interpret IoT as a well-defined scheme of interconnected computers, digital and mechanical devices that possess the ability to transmit data over a defined network without any human involvement at any level. IoT is a centralized digital platform that combines data collected based on advanced technologies, such as RFID tags, bar codes, wireless sensors (WS), cloud computing, and artificial intelligence (AI). It is a system that collects data, analyzes them, and makes timely decisions based on the obtained inputs [32,33]. In particular, in FMCG supply chains, the IoT platform is based on the interconnection of all participants in the supply chain, production, and service processes through sensors located on the product itself or its packaging [12]. IoT technology for linking production and service process data enables the FMCG supply chain to more efficiently control the entire supply system, from the manufacturer to the retailer and the final consumer, allowing for high transparency, sustainability, and product monitoring for better safety and security [34].

There are several research studies conducted that indicate the importance of introducing integrated IoT platforms in supply chain management. These are, first of all, security, safety, and sustainability

of product placement [12], higher profitability and productivity of the chain [17], transparency and visibility of data for all participants in the chain [7], visibility of stocks and real-time data collection [34], increased transparency of physical distribution (transport conditions, destinations, etc.) available to the entire supply chain [34], more efficient control of storage conditions [31], timely response to market and final consumer needs [35]. Such networking and data storage based on monitoring the activities of each manufacturer, distributor, wholesaler, and retailer, as well as monitoring each purchase and consumer, leaves room for FMCG supply chain management to make timely, business efficient and effective performance decisions about the needs of particular markets. Figure 1 shows a detailed graphical representation of the integrated IoT platform for FMCG supply chain management.

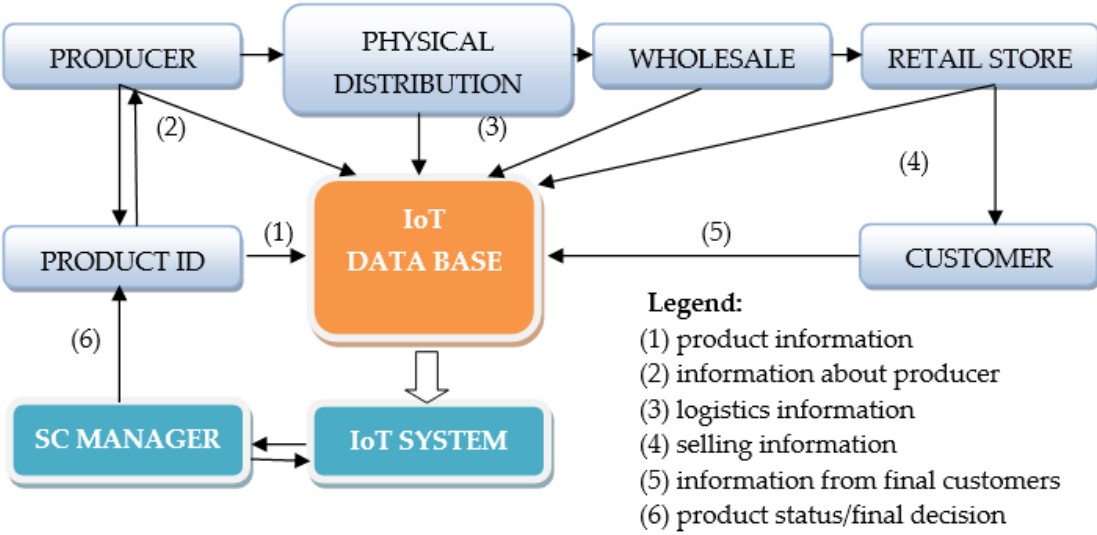

**Figure 1.** Internet of Things (IoT) based supply chain.

Figure 1 systematizes the six phases of product monitoring along the entire supply chain through IoT technology. Phase I: data on each product (composition, shelf life, warranty period, etc.) via RFID tags, sensors, or bar codes, through a reader, is stored in a central database. Phase II: manufacturers will enter all information about raw materials, method of production, time and place of production, etc. Phase III: distributors provide information on transport conditions, delivery speed, storage conditions, and how to store the product. Phase IV: Wholesalers and retailers place information on sales volume, available quantities, and assortment, as well as demand volume and return rate. Phase V: the IoT database also stores information from the final consumers themselves, which detects the use-value of the product, quality, choice, complaints, etc. Phase VI: The IoT system analyzes the information from the database and presents it to the supply chain management, which makes the final decision on the status of the product, placement method, eliminating critical places in supply, more efficiently meeting the needs of markets and consumers, etc. All information is shared through all stages of the supply chain. Such a digitalized system ensures the accuracy and timely exchange of information, from upstream to downstream participants in the chain, and vice versa, with minimal participation of the human factor, which makes it suitable during large market fluctuations such as the COVID-19 pandemic.

### 2.3. The Setbacks to IoT Implementation in the FMCG Supply Chain

Despite the undeniable advantages, IoT technology is not widely represented everywhere. Recent studies indicate certain obstacles to the application of IoT. Lee and Lee [36] cite problems with data comprehension. The data contained in the IoT database consists not only of traditional discrete data but also of streaming data generated by digital sensors located on products, services, vehicles, etc. Analysis of such data requires advanced computing devices, mathematical models, as well as highly qualified and trained analysts. The results of the study by Wang et al. [37], Gubbi et al. [18],

and Lee et al. [36] show that about 70% of imported products, services, and users have vulnerabilities due to a lack of secure web interfaces, inadequate software protection and insufficient authorization, which opens up security problems for IoT users. Končar et al. [17] and Granjal et al. [38] point to issues related to the high initial costs of installing an IoT platform. Bedekar [39] considers the crucial problem to be insufficient training of all participants in the supply chain to apply advanced technologies and the lack of skilled labor, especially in developing countries and transition countries. Data privacy in their research was defined by Hameed et al. [40] as the inability of IoT application users to protect their data related to their movement, consumption, needs, habits, and interaction with other people. According to the same author, another threat is localization and tracking, because the IoT platform tries to determine and record the location of a person, product, and service through time and space. Whitmore et al. [41] emphasize that the implementation of IoT in supply chains requires changes in the organization of the chain itself, as well as changes in the business model, for which participants of the chain are not ready. Botta et al. [42] and Luthra et al. [43] consider the underdeveloped existing infrastructure, especially in transition countries, to be a massive obstacle to the implementation of IoT in supply chains, which is not able to effectively support and successfully manage interconnected products, equipment, and devices. Al-Fugaha et al. [44] warn about the problems of standardization, certification, and permits, while Luthra et al. [43] and Bedekar [38] consider the lack of internet connection, frequent signal interruptions, low use of WiFi connection, and so forth, to be a significant problem, especially in rural areas. Finally, Končar et al. [17], Andjelković and Radosavljević [45] in their research also mention the lack of awareness, both among the participants in the supply chain and final consumers and the market, about the need to digitalize business operations based on IoT. The identified setbacks to IoT implementation are summarized in Table 1.

**Table 1.** Setbacks to implementation of IoT in fast-moving-consumer-goods (FMCG) supply chains.

| No | Setback/ Indicator | Description | Sources |
|:---:|:---:|:---:|:---|
| 1 | Data analysis and comprehension | Analysis of streaming data generated by digital sensors requires advanced computing devices, knowledge of mathematical models, as well as highly qualified and trained analysts | Lee and Lee [36] |
| 2 | User safety | User vulnerabilities due to lack of secure web interfaces, inadequate software protection, and insufficient authorization | Wang et al. [37] Gubbi et al. [18] Lee et al. [36] |
| 3 | Lack of financial resources | The implementation of IoT platforms requires substantial initial investments and enormous initial system maintenance cost | Končar et al. [17] Granjal et al. [38] |
| 4 | Lack of skilled workers | Insufficient training of all participants in the supply chain to apply advanced technologies and lack of skilled labor | Bedekar [39] |
| 5 | Privacy protection | Inability of IoT application users to protect their data related to their movement, consumption, needs, habits, and interactions | Hameed et al. [40] |
| 6 | Changing the business model | The implementation of IoT requires changes in the organization of the chain itself, for which participants are often not ready | Whitmore et al. [41] |
| 7 | Underdeveloped infrastructure | Infrastructure that is not able to effectively support and successfully manage interconnected products, equipment, and devices | Botta et al. [42] Luthra et al. [43] |
| 8 | Standardization | Problems with the introduction of new standards, certificates and permits for the implementation of IoT | Al-Fugaha et al. [44] |
| 9 | Lack of internet connection | Lack of internet connection, frequent signal interruptions, low use of WiFi connection, etc. | Luthra et al. [43] Bedekar [39] |
| 10 | Lack of awareness | Existence of an unclear need to digitalize IoT-based supply chain business operations. | Končar et al. [17] Anđelković and Radosavljević [45] |

In the WB countries, in the period before the COVID-19 pandemic, the application of IoT systems in supply chains was at a very low level. Moreover, only a few academic studies have indicated the need for digitalization. FMCG supply chain management is leaning more towards the traditional

management approach where traditional retail facilities, physical contact with consumers, analog data entry, etc., dominate. In the first quarter of 2020, according to the Eurostat report [46], the WB countries achieved significant GDP growth rates. For example, Serbia +5% (1st place in Europe), Montenegro +2.7%, Croatia +0.3%, North Macedonia +0.2%. However, the COVID-19 pandemic significantly changed economic and business activities in the second quarter of 2020. According to the European Commission report, the expected decline in GDP for the WB region ranges from −4.1% (Serbia) to −11% (Croatia) [47]. Tourism, hotels, public transport, and retail sectors suffered the biggest shockwaves. Measures of social distancing, work from home, increased protocols on the safety and health of products and people, and the accumulation of basic foodstuffs have significantly burdened supply chains. The demand for products necessary in self-isolation, such as food +55.8%, medicines +88.2%, and hygiene supplies +29.2%, is growing fast. On the other hand, in other categories, sales fell by 18.1%, namely: footwear and clothing 35%, electronics 20.9%, IT devices 22.7, etc.

The most significant characteristic of supply chains is an appropriate allocation of resources in the sense that products are made in sufficient quantities and placed on the target market segment at the right time. Market fluctuations have further fueled the need to build a more flexible supply chain that would be sustainable in times of considerable disturbances such as the COVID-19 pandemic. In this regard, there is a need to define the reasons and setbacks that limit the digitalization of the FMCG supply chain in the WB region. Based on the presented theoretical aspects, the main objective of this paper is to determine the reasons for the low rate of application of digitalization in FMCG supply chains, through analysis and testing of the real impact of identified setbacks (Table 1) for IoT implementation. For the research, FMCG supply chains of the WB region were selected due to significant differences in the level of economic development among chosen countries, different demographic potential, and different approaches to sustainability issues (e.g., EU/non-EU countries). As such, the sample is representative and, at the same time, convenient for comparative analysis. Additionally, the European Commission in its Strategy for the Danube Region identified better connectivity between the WB countries through the digitalization of their business processes as one of its priority areas [48].

## 3. Methodology

### 3.1. Aim and Hypotheses

Based on the analyzed theoretical views and the results of research of various studies on the importance of IoT implementation in FMCG supply chains, setbacks for its successful application have been identified [17,18,36–45]. In this regard, the primary research goal is to clearly define the impact of the identified setbacks to the digitalization of business processes and the sustainability of the FMCG supply chain based on the implementation of the Internet of Things in the WB region. The set goal of the research was operationalized through three basic and several supporting hypotheses.

A review of the literature [17,18,36–45] identified ten setbacks for the introduction and application of IoT in supply chains. As different intensity of significance and connection with IoT implementation can be expected for these setbacks, research hypothesis $H_1$ states that the identified setbacks are statistically significant in predicting the delay of IoT implementation in FMCG supply chains in the WB region. To adequately test the first research hypothesis, it should be supported by ten auxiliary research hypotheses with each of the defined setbacks individually. $H_1$ (a)—Analysis and comprehension of data is statistically significant in predicting delay of IoT implementation in FMCG supply chains in the WB region, $H_1$ (b)—User safety in a statistically significant way predicts delay of IoT implementation in FMCG supply chains in the WB region, $H_1$ (c)—Lack of financial resources is statistically significant in predicting delay of IoT implementation in FMCG supply chains in the WB region, $H_1$ (d)—Lack of skilled workers is statistically significant in predicting delay of IoT implementation in FMCG supply chains in the WB region, $H_1$ (e)—Privacy protection in a statistically significant way predicts delay of IoT implementation in FMCG supply chains in the WB region, $H_1$ (f)—Change of business model in a statistically significant way predicts delay of IoT implementation in FMCG supply chains

in the WB region, $H_1$ (g)—Underdeveloped infrastructure in a statistically significant way predicts delay of IoT implementation in FMCG supply chains in the WB region, $H_1$ (h)—Standardization in a statistically significant way predicts implementation delay of IoT in FMCG supply chains in the WB region, $H_1$ (i)—Lack of internet connection in a statistically significant way predicts delay in IoT implementation in FMCG supply chains in WB, $H_1$ (j)—Lack of awareness of the need for digitalization in a statistically significant way predicts delay of IoT implementation in FMCG supply chains in the WB region.

Some studies [12,14,17–19,32,35,49] show that there are significant differences between manufacturers, distributors, wholesalers, and retailers as participants in the FMCG supply chain, in terms of acceptance of advanced technology, modern logistics concepts, attitudes towards consumers and the environment, social responsibility, etc. In this context, under the set research goal, as a separate research hypothesis, it is necessary to test the existence of differences between obstacles to the implementation of IoT platforms in the manufacturing, transport, wholesale and retail FMCG sector. The second research hypothesis $H_2$ reads: setbacks to IoT implementation differ in a statistically significant way among the main participants in the FMCG supply chain in the WB region.

The WB region consists of six countries (Croatia, Serbia, Northern Macedonia, Bosnia and Herzegovina, Montenegro, and Albania) with significant differences in terms of levels of economic development, level of development of FMCG supply chains, growth of the retail network, demographic characteristics, market needs, and consumers, etc. Furthermore, some countries are members of the EU (Croatia) or in the process of pre-accession negotiations for EU accession (Serbia) and have harmonized laws with EU Directives, compared to the rest of the Western Balkans. As the research focus of this paper is on the WB region consisting of independent national markets, it is necessary to test whether differences in setbacks to IoT implementation in FMCG supply chains are related to differences between WB countries. The research by Končar et al. from 2019 partially confirms these assumptions [17]. In this regard, the latest research hypothesis $H_3$ states that the differences in the setbacks to the implementation of IoT in the FMCG supply chain are statistically significantly related to the differences between the observed countries in the WB region.

In addition to the scientific, the set hypotheses also have a practical contribution. Based on the defined setbacks and differences, a set of measures and incentives that the competent institutions and the management of the FMCG supply chain can take to strengthen the digitalization process is recommended.

### 3.2. Research Variables

Hypotheses testing includes dependent and multiple independent variables. Setbacks for the implementation of IoT in supply chains were chosen as independent variables of interval type of measurement. These are data analysis and comprehension, user security, lack of financial resources, lack of skilled workers, privacy protection, change of business model, underdeveloped infrastructure, standardization, lack of internet connection, and lack of awareness of the need for digitalization [17,18,36–45]. The dependent variable is the delay in IoT implementation in the FMCG supply chains of the WB region, which is estimated based on the impact of the independent variables. Participants in the FMCG supply chain (production, physical distribution, wholesale, and retail) and the analyzed WB countries (Croatia, Serbia, Northern Macedonia, Bosnia and Herzegovina, and Montenegro) were selected as independent grouping variables.

### 3.3. Research Sample

The research was conducted on a sample of 209 middle and lower-level managers in FMCG supply chains in the WB countries. The given sample belongs to the category of large statistical samples, and representativeness is ensured by the fact that the analyzed respondents are engaged in positions and work tasks where they directly face setbacks to the implementation of IoT. In terms of sample structure, the number of respondents is uniform across countries and participants in the FMCG supply

chain, as well as basic demographic categories. The detailed structure of the research sample is shown in Table 2.

**Table 2.** Research sample structure.

| Gender | n | Structure (%) | Age | n | Structure (%) |
|---|---|---|---|---|---|
| Male | 121 | 57.9% | up to 30 | 34 | 16.3% |
| Female | 88 | 42.1% | 30–40 | 51 | 24.4% |
| | | | 40–50 | 78 | 37.3% |
| | | | over 50 | 46 | 22.0% |
| **WB Countries** | **n** | **Structure (%)** | **FMCG Supply Chain Sector** | **n** | **Structure (%)** |
| Crotia | 42 | 20.1% | Production | 49 | 23.4% |
| Serbia | 44 | 21.1% | Physical Distribution | 52 | 24.9% |
| Bosnia and Hercegovina | 40 | 19.1% | Wholesale | 51 | 24.4% |
| North Macedonia | 42 | 20.1% | Retail | 57 | 27.3% |
| Montenegro | 41 | 19.6% | | | |

Source: Author's calculation.

### 3.4. Research Procedure and Data Analysis

The research was conducted in the period from May to July 2020, electronically based on an anonymous questionnaire. The questionnaire was created based on the questionnaires and indicators used by Luthra et al. in their 2018 research [43], that is, by Končar et al. [17] in their 2020 research. After general demographic information (gender, age) and affiliation information (FMCG supply chain sector, WB country), respondents ranked the ten setbacks offered for IoT implementation in FMCG supply chains. Each of the obstacles was examined based on three items that were ranked by the standard Likert scale (0-no significance; 5-very high significance). The dependent variable (IoT implementation) was also examined based on three Likert-type items.

The research included managers from the largest FMCG supply chains operating in the WB region, such as Delta Transport Logistics, Lidl, Univerexport, Pluto Logistics, Ralu Logistics, DIS. The survey was conducted in five WB countries with which there is good institutional cooperation (Croatia-Cro, Serbia-Srb, Bosnia and Herzegovina-B&H, North Macedonia-Mcd, Montenegro-Mng). The response rate of the respondents was 64%.

The reliability of the selected scales was tested using the internal consistency coefficient of Cronbach's alpha, whose value for all items is above 0.650, which shows that the questions chosen describe the same problem, i.e., they can be used to assess the attitudes and opinions of FMCG supply chain managers. Statistically significant deviations are not observed when applying the coefficient Skewness (distribution symmetry) and Kurtosis (distribution of results), which further confirms the reliability and correctness of the selected scales. Table 3 presents detailed values of the mentioned coefficients according to the analyzed indicators.

The collected data were sorted and processed using the statistical package SPSS 20. Appropriate statistical methods were used to present the research results and test the set research hypotheses such as descriptive statistics (description of primary statistical indicators of the sample), one-way analysis of variance, Pearson correlation coefficient, and multiple regression analysis (hypotheses testing).

**Table 3.** Values of Cronbach Alpha, Skewness, and Kurtosis coefficient.

| Indicators | Cronbach's Alpha | Skewness | Kurtosis |
|---|---|---|---|
| Data analysis and comprehension | 0.911 | −0.069 | −1.040 |
| User safety | 0.728 | −0.108 | −1.017 |
| Lack of financial resources | 0.839 | 0.341 | −0.885 |
| Lack of skilled workers | 0.701 | −0.327 | −1.273 |
| Privacy protection | 0.672 | −0.212 | −1.248 |
| Changing the business model | 0.755 | 0.057 | −1.538 |
| Underdeveloped infrastructure | 0.932 | −0.319 | −0.545 |
| Standardization | 0.811 | 0.273 | −1.143 |
| Lack of internet connection | 0.683 | 0.322 | −1.300 |
| Lack of awareness | 0.847 | 0.224 | −1.655 |

Source: Author's calculation.

## 4. Results

The obtained average responses of the respondents, that is, the degree of agreement with the claims that the identified setbacks limit the implementation of IoT in FMCG supply chains in the WB region are presented in Table 4. The presented ranks by participants in the supply chain (columns 3, 4, 5, 6) represent the obtained average scores based on three items from the questionnaire for each listed indicator. The last column (column 7) is the average score of the given indicator for all participants in the chain.

The biggest obstacle to the process of digitalization of the FMCG supply chain in the WB region is the lack of financial resources and high initial costs of IoT implementation (M = 4.21). This result is expected, given the fact that the COVID-19 pandemic has caused an unforeseeable economic crisis, and supply chain participants are thinking more about saving as a recovery measure than investing in advanced technologies that will bring benefit after the high initial investment. High average scores are recorded in underdeveloped infrastructure (M = 3.84) and lack of skilled workers (M = 3.78). On the other hand, the lack of internet connection (M = 2.60) is the least setback, which indicates the relatively good availability of the Internet in rural parts of the WB. Observed individually by participants in the FMCG supply chain, the biggest obstacle for producers and wholesalers is the lack of financial resources (M = 4.41; M = 4.11), for participants in physical distribution, it is data privacy (M = 4.23), while retailers see problems related to the analysis and processing of collected data (M = 4.21) as the biggest challenge for the implementation of IoT. Table 5 summarizes the most significant demographic indicators.

The highest degree of agreement of FMCG supply chain participants is grouped around the indicator related to underdeveloped infrastructure (SD = 0.64819). Respondents agree that WB supply chains do not have state-of-the-art equipment and technology capable of supporting IoT-based data networking. Respondents also show a high degree of agreement with the lack of financial resources (SD = 0.8425), internet connection (SD = 0.8675) and data understanding (SD = 0.8715). On the other hand, respondents least agree about the willingness to change the business model and the way of organizing business with the implementation of IoT (SD = 1.1724). Respondents mostly disagree that FMCG supply chains lack quality and highly qualified personnel capable of implementing and managing IoT (SD = 1.0027).

For broader insight into the impact of these setbacks to the digitalization of the FMCG supply chain, it is necessary to define which of the analyzed setbacks act together, that is, test their correlation to determine which setbacks have similar coefficients. The Correlation Matrix (Table 6) presents detailed correlations between indicators.

In the presented matrix, the most significant degree of mutual correlation is reflected in the lack of qualified workers, lack of financial resources, underdeveloped infrastructure, and analysis and comprehension of data indicators. The obtained results show that the participants in the FMCG supply chain, who state the high costs of IoT implementation as the most impeding setback, most often state

the lack of staff (r = 0.77) and underdeveloped infrastructure as well (r = 0.91). Significant correlations can be seen between the lack of skilled workers and underdeveloped infrastructure (r = 0.86) and the setback to analysis and comprehension of data, which is most correlated with user safety (r = 0.84) and underdeveloped infrastructure (r = 0.78). It is interesting to note that a significant correlation is also seen in the indicators of change of the business model with which respondents most often state underdeveloped infrastructure (r = 0.61), weak internet connection (r = 0.75), and lack of awareness of the need for digitalization (r = 0.54). Statistically, significant correlations are not observed between the indicators of standardization, lack of internet connection, and lack of awareness among the participants of the FMCG supply chain about the need to digitalize data based on IoT.

**Table 4.** Review of ranking results.

| Ord. No. | Indicators (Setbacks) | Production | Physical Distribution | Wholesale | Retail | Mean (M) |
|---|---|---|---|---|---|---|
| 1 | 2 | 3 | 4 | 5 | 6 | 7 |
| 1 | Data analysis and comprehension | 4.11 | 2.83 | 3.72 | 4.21 | **3.72** |
| 2 | User safety | 3.87 | 3.14 | 3.21 | 3.12 | **3.34** |
| 3 | Lack of financial resources | 4.41 | 4.13 | 4.25 | 4.05 | **4.21** |
| 4 | Lack of skilled workers | 3.96 | 3.35 | 4.11 | 3.70 | **3.78** |
| 5 | Privacy protection | 3.15 | 4.23 | 3.77 | 3.41 | **3.64** |
| 6 | Changing the business model | 4.01 | 3.27 | 2.88 | 2.96 | **3.28** |
| 7 | Underdeveloped infrastructure | 3.91 | 3.52 | 4.09 | 3.83 | **3.84** |
| 8 | Standardization | 3.86 | 3.28 | 2.62 | 3.40 | **3.29** |
| 9 | Lack of internet connection | 2.74 | 2.67 | 2.11 | 2.89 | **2.60** |
| 10 | Lack of awareness | 3.14 | 2.62 | 3.04 | 3.73 | **3.13** |

Source: Author's calculation.

**Table 5.** Descriptive statistics for selected indicators.

| Ord. No. | Indicators (Setbacks) | Min | Max. | Mean (M) | Standard Error (SE) | Standard Deviation (SD) |
|---|---|---|---|---|---|---|
| 1 | 2 | 3 | 4 | 5 | 6 | 7 |
| 1 | Data analysis and comprehension | 2.00 | 5.00 | 3.72 | 0.0603 | 0.8715 |
| 2 | User safety | 1.00 | 4.00 | 3.34 | 0.0649 | 0.9381 |
| 3 | Lack of financial resources | 2.00 | 5.00 | 4.21 | 0.0583 | 0.8425 |
| 4 | Lack of skilled workers | 1.00 | 5.00 | 3.78 | 0.0694 | 1.0027 |
| 5 | Privacy protection | 1.00 | 5.00 | 3.64 | 0.0811 | 1.1724 |
| 6 | Changing the business model | 1.00 | 5.00 | 3.28 | 0.0851 | 1.2308 |
| 7 | Underdeveloped infrastructure | 2.00 | 5.00 | 3.84 | 0.0448 | 0.6481 |
| 8 | Standardization | 1.00 | 5.00 | 3.29 | 0.0880 | 1.2728 |
| 9 | Lack of internet connection | 1.00 | 4.00 | 2.60 | 0.0600 | 0.8675 |
| 10 | Lack of awareness | 1.00 | 4.00 | 3.13 | 0.0638 | 0.9224 |

Source: Author's calculation.

Testing of the first research hypothesis $H_1$ requires examination of the group of supporting hypotheses $H_1$ (a)–$H_1$ (j). Multiple regression analysis was applied to test the connection between the mentioned group of indicators and the concept of digitalization based on IoT. Testing will first be conducted on the total sample of surveyed FMCG supply chain managers, and then for each setback individually. The Enter method, in which all independent variables (setbacks) were included together to predict the dependent variable (inability to implement IoT) was used for testing. The obtained results indicate that the regression model is statistically significant (F (200; 9) = 2.50, $p < 0.01$), which means that the set of examined setbacks statistically significantly predicts the implementation of the digital strategy based on IoT in FMCG supply chains in the WB region. It describes 66.2% of the variance of the criteria. Besides the total contribution of the set of indicators, the contribution of individual predictors is presented in Table 7.

**Table 6.** Correlation matrix.

| | | 1 | 2 | 3 | 4 | 5 | 6 | 7 | 8 | 9 | 10 |
|---|---|---|---|---|---|---|---|---|---|---|---|
| 1 | **Data analysis and comprehension** | 1 | | | | | | | | | |
| 2 | **User safety** | 0.84 ** | 1 | | | | | | | | |
| 3 | **Lack of financial resources** | 0.89 | 0.73 | 1 | | | | | | | |
| 4 | **Lack of skilled workers** | 0.60 * | 0.36 | 0.77 ** | 1 | | | | | | |
| 5 | **Privacy protection** | 0.59 * | 0.75 ** | 0.49 | 0.59 | 1 | | | | | |
| 6 | **Changing the business model** | 0.73 | 0.83 | 0.35 | 0.47 | 0.53 | 1 | | | | |
| 7 | **Underdeveloped infrastructure** | 0.78 ** | −0.87 | 0.91 ** | 0.86 ** | −0.25 | 0.61 * | 1 | | | |
| 8 | **Standardization** | 0.34 | 0.51 | 0.64 | 0.66 * | 0.54 | 0.52 | 0.72 * | 1 | | |
| 9 | **Lack of internet connection** | 0.48 | 0.61 | 0.57 | 0.63 | −0.67 | 0.75 * | 0.58 | 0.46 | 1 | |
| 10 | **Lack of awareness** | 0.54 | 0.56 | 0.44 | 0.49 | −0.55 | 0.54 * | 0.60 | 0.56 | 0.62 | 1 |

Note: ** Correlation is significant at the level 1%; * Correlation is significant at the level 5%. Source: Author's calculation.

**Table 7.** Testing the individual indicators contribution.

| | Stand. Coefficient | | *t* | *Sig.* |
| --- | --- | --- | --- | --- |
| | **Beta** | **St. Error** | | |
| (const.) | 0.69 | 1.163 | 3.577 | 0.000 |
| Data analysis and comprehension | 0.462 ** | 0.362 | 0.528 | 0.045 |
| User safety | 0.865 | 0.476 | 0.813 | 0.183 |
| Lack of financial resources | 0.776 ** | 0.403 | 0.620 | 0.000 |
| Lack of skilled workers | 0.683 ** | 0.612 | 1.032 | 0.004 |
| Privacy protection | 0.492 * | 0,781 | 0.297 | 0.016 |
| Changing the business model | 0.518 * | 0.576 | 1.251 | 0.047 |
| Underdeveloped infrastructure | 0.559 ** | 0.391 | 0.405 | 0.004 |
| Standardization | 0.491 | 0.584 | 0.624 | 0.063 |
| Lack of internet connection | 0.517 | 0.374 | 0.416 | 0.084 |
| Lack of awareness | 0.255 | 0.572 | 0.734 | 0.191 |

Note: ** Correlation is significant at the level 1%; * Correlation is significant at the level 5%. Source: Author's calculation.

Table 7 shows that the lack of financial resources (B = 0.776; $p < 0.01$), underdeveloped infrastructure (B = 0.559; $p < 0.01$), lack of skilled workers (B = 0.683; $p < 0.01$), and analysis and comprehension of data are statistically significant in predicting delay in the implementation of IoT in FMCG supply chains. The obtained result coincides with the findings of previous studies [17,36–39,42,43] which see the biggest problem of digitalization of supply chains in high investments, lack of trained workers for successful implementation and maintenance of IoT, and lack of modern infrastructure to support IoT in terms of modern IT equipment, smart devices, artificial intelligence, etc. As a result of the last two setbacks, a problem related to the analysis and comprehension of the collected data arises, that is, the timely reaction of the chain based on such data. These setbacks are especially emphasized in underdeveloped and transition countries that are in the early stages of the digitalization process. The correlation between these setbacks and IoT implementation is positive and statistically significant, which means that with increasing setback intensity, the unwillingness of FMCG supply chain participants to implement IoT increases, and vice versa. Respondents put less importance on the privacy protection (B = 0.492, $p < 0.05$) and readiness to change the business model (B = 0.518, $p < 0.05$). Other analyzed indicators do not have statistical significance for predicting the implementation of IoT. Based on the conducted testing, we can conclude that the supporting hypotheses H$_1$ (a), H$_1$ (c), H$_1$ (d), H$_1$ (e), and H$_1$ (g) are accepted and that setbacks such as lack of financial resources, underdeveloped infrastructure, lack of skilled workers, problems with data analysis and comprehension, privacy protection and the need to change the business model and the way the chain operates, predict the delay of the FMCG supply chain digitalization process in the WB region. The remaining supporting hypotheses H$_1$ (b), H$_1$ (h), H$_1$ (i), and H$_1$ (j) are rejected, which means that user safety, the standardization problem, weak internet connection, and lack of awareness and readiness for digitalization have no impact on the application of IoT. These results suggest that the first research hypothesis H$_1$ is partially accepted.

As the focus of the conducted research is on FMCG supply chain participants in the WB region, it is necessary to determine whether the differences that occur in setbacks to IoT implementation are statistically significantly related to differences between the production, physical distribution, wholesale and retail sectors. To test the second research hypothesis H$_2$, which examines this correlation, multiple regression analysis was applied. Table 8 shows detailed test results by supply chain participants and analyzed indicators.

When looking at the contributions of individual indicators, it can be seen that the description of the criterion variance (delay in IoT implementation) is mostly contributed by data analysis and comprehension, lack of financial resources, lack of skilled workers, and underdeveloped infrastructure indicators. Observed individually by participants in the FMCG supply chain with manufacturers, the statistical significance is recorded for the setback related to changes in the business model ($\beta = 0.747$,

$p < 0.01$). In the physical distribution sector, this is user security ($\beta = 0.929$, $p < 0.05$) and privacy protection ($\beta = 0.703$, $p < 0.01$). For wholesalers, a significant obstacle is the standardization of business processes ($\beta = 0.527$, $p < 0.05$), while retailers emphasize the protection of privacy ($\beta = 0.704$, $p < 0.01$). Other analyzed indicators do not have a statistically significant impact. The presented results suggest that the obtained regression equations are statistically significant for each of the observed participants in the FMCG supply chain ($p$: (F (50, 9) = 2.88, $p < 0.05$; phd: (F (50, 9) = 2.88, $p < 0.01$; v: F (50, 9) = 2.88, $p < 0.01$; r: (F (50, 9) = 2.88, $p < 0.01$). We can conclude that the differences between the sectors of production, physical distribution, wholesale and retail are statistically significantly related to differences in setbacks for digitalization processes in FMCG supply chains. In other words, setbacks to IoT implementation differ statistically significantly among major FMCG supply chain participants in the WB region, thus making the second research hypothesis $H_2$ acceptable. This confirms the results of some previous studies on the existence of significant differences between supply chain participants in terms of acceptance of advanced technologies [12,14,17–19,32,35,49].

**Table 8.** Testing the individual indicators contribution by participants of FMCG supply chain.

| | Production (p) | | Physichal Distribution (phd) | | Wholesale (w) | | Retail (r) | |
|---|---|---|---|---|---|---|---|---|
| | **Beta** | **Sig.** | **Beta** | **Sig.** | **Beta** | **Sig.** | **Beta** | **Sig.** |
| (const.) | 0.81 | 0.017 | 0.74 | 0.003 | 0.62 | 0.000 | 0.74 | 0.001 |
| Data analysis and comprehension | 0.573 ** | 0.000 | 0.512 | 0.073 | 0.662 * | 0.033 | 0.795 ** | 0.002 |
| User safety | −0.608 | 0.055 | 0.929 * | 0.012 | 0.737 | −0.081 | 0.623 | 0.557 |
| Lack of financial resources | 0.490 ** | 0.001 | 0.627 ** | 0.000 | 0.914 ** | 0.000 | 0.769 * | 0.041 |
| Lack of skilled workers | 0.223 * | 0.042 | 0.036 | 0.790 | 0.228 ** | 0.003 | 0.095 ** | 0.003 |
| Privacy protection | 0.582 | 0.184 | 0.703 ** | 0.002 | - 0.653 | 0,694 | 0.704 ** | 0.008 |
| Changing the business model | 0.747 ** | 0.006 | 0.626 | 0.088 | 0.854 | 0.142 | 0.791 | 0.100 |
| Underdeveloped infrastructure | 0.711 * | 0.024 | 0.559 ** | 0.001 | 0.605 ** | 0.007 | 0.622 * | 0.013 |
| Standardization | −0.891 | 0.057 | 0.782 | −0.247 | 0.527 * | 0.046 | 0.303 | 0.114 |
| Lack of internet connection | −0.522 | 0.315 | −0.358 | 0.681 | 0.694 | 0.067 | 0.433 | 0.323 |
| Lack of awareness | 0.671 | 0.066 | 0.581 | 0.054 | 0.481 | 0.112 | 0.191 | 0.074 |

Note: ** Correlation is significant at the level 1%; * Correlation is significant at the level 5%. Source: Author's calculation.

The last research hypothesis $H_3$ examines whether the differences in the setbacks to the implementation of IoT in the FMCG supply chain are statistically significantly related to the differences between the chosen countries in the WB region. Hypothesis $H_3$ determines whether different setbacks dominate the individual national markets (Cro, Srb, B&H, Mcd, Mng) which delay the implementation of IoT. The correlation between defined setbacks and delays in IoT implementation was tested by a series of Pearson correlations, separately for each analyzed WB country.

Table 9 shows that similar setbacks to the digitalization of business and service processes in supply chains are statistically significant in all analyzed WB countries. These are problems related to data analysis, lack of financial resources, lack of skilled workers, underdeveloped infrastructure, privacy protection, and changes in the way of organization and business model. The remaining indicators in no WB country have statistical significance for delaying IoT implementation. Although some correlations are of lower intensity, and small differences between indicators are noticeable, no statistically significant discrepancies can be identified among WB countries. That leads to the conclusion that the delay in the process of IoT implementation in the FMCG supply chain is related to setbacks that are equally manifested in each of the selected countries, thus rejecting the last research hypothesis $H_3$. In other words, when it comes to FMCG supply chains, the WB market can be viewed as a single entity. That is perhaps the expected result, given that the supply chains operating in the WB countries are mainly regional in nature and directed at supplying the entire WB market.

**Table 9.** Correlation of setbacks and implementation of IoT by Western Balkans (WB) countries.

| Implementation of IoT | Analysis and Data Comprehension | User Safety | Lack of Financial Resources | Lack of Skilled Workers | Privacy Protection | Change of the Business Model | Underdeveloped Infrastructure | Standardization | Lack of Internet Connection | Lack of Awareness |
|---|---|---|---|---|---|---|---|---|---|---|
| **Croatia** | | | | | | | | | | |
| Correlation coefficient (r) | 0.722 ** | 0.306 | 0.689 ** | 0.397 ** | 0.687 * | 0.514 * | 0.784 * | 0.475 | 0.655 | 0.475 |
| Significance | 0.000 | 0.214 | 0.000 | 0.008 | 0.025 | 0.047 | 0.001 | 0.142 | 0.055 | 0.100 |
| Number | 42 | 42 | 42 | 42 | 42 | 42 | 42 | 42 | 42 | 42 |
| **Serbia** | | | | | | | | | | |
| Correlation coefficient (r) | 0.641 ** | 0.156 | 0.763 ** | 0.624 ** | 0.867 ** | 0.792 * | 0.386 ** | 0.861 | 0.660 | 0.704 |
| Significance | 0.008 | 0.074 | 0.000 | 0.001 | 0.003 | 0.035 | 0.004 | 0.110 | 0.081 | 0.817 |
| Number | 44 | 44 | 44 | 44 | 44 | 44 | 44 | 44 | 44 | 44 |
| **Bosnia and Herzegovina** | | | | | | | | | | |
| Correlation coefficient(r) | 0.662 ** | 0.381 | 0.782 ** | 0.838 ** | 0.630 * | 0.872 * | 0.746 ** | 0.482 | 0.516 | 0.441 |
| Significance | 0.000 | 0.071 | 0.003 | 0.000 | 0.021 | 0.000 | 0.001 | 0.066 | 0.141 | 0.091 |
| Number | 40 | 40 | 40 | 40 | 40 | 40 | 40 | 40 | 40 | 40 |
| Montenegro | | | | | | | | | | |
| Correlation coefficient (r) | 0.487 ** | 0.142 | 0.597 ** | 0.609 * | 0.490 * | 0.717 * | 0.496 ** | 0.215 | 0.,234 | 0.500 |
| Significance | 0.001 | 0.100 | 0.003 | 0.014 | 0.044 | 0.000 | 0.000 | 0.107 | 0.059 | 0.084 |
| Number | 42 | 42 | 42 | 42 | 42 | 42 | 42 | 42 | 42 | 42 |
| **Northern Macedonia** | | | | | | | | | | |
| Correlation coefficient (r) | 0.765 * | 0.236 | 0.650 ** | 0.788 ** | 0.582 * | 0.560 * | 0.833 ** | 0.406 | 0.610 | 0.320 |
| Significance | 0.031 | 0.426 | 0.001 | 0.000 | 0.040 | 0.033 | 0.001 | 0.202 | 0.069 | 0.110 |
| Number | 41 | 41 | 41 | 41 | 41 | 41 | 41 | 41 | 41 | 41 |

Note: ** Correlation is significant at the level 1%; * Correlation is significant at the level 5%. Source: Author's calculation.

## 5. Discussion

*5.1. Defining a Model for Predicting the Significance of Setbacks to Delaying IoT Implementation*

The conducted research and the obtained results have identified significant obstacles that delay the process of digitalization of the FMCG supply chain based on IoT. Due to this reason, FMCG supply chains in the WB market are becoming less flexible, transparent, and efficient. The COVID-19 pandemic further deepened these problems and called into question the sustainability of the supply chain in a period of sudden market shocks and large fluctuations on the demand side. These are situations in which a sustainable, fully digitalized, less human-dependant system needs to be built. Given that delaying the IoT implementation process in FMCG supply chains can be treated as a problem of one dependent and several independent variables, this is a suitable situation for data analysis using a multiple linear regression model:

$$y = 0.69 + 0.462 \times x_1 + 0.776 \times x_3 + 0.683x_4 + 0.492x_5 + 0.518x_6 + 0.559x_7 \tag{1}$$

In the given model, only setbacks that have a statistically significant impact on the delay of the IoT implementation process in FMCG supply chains ($y$) are presented. These are $x_1$—analysis and comprehension of data, $x_3$—lack of financial resources, $x_4$—lack of skilled workers, $x_5$—privacy protection, $x_6$—change of business model, and $x_7$—underdeveloped infrastructure.

The model can serve as a primary guide for the competent institutions, economic policymakers, and FMCG supply chain management in WB, to point out obstacles that need to be minimized to encourage IoT implementation and thus make the supply chain more flexible to the challenges caused by the COVID-19 pandemic. Digitalization would build a supply system in the WB region to compensate for the problems imposed by lockdown policy, social distancing, and staff shortages due to illness and self-isolation while responding more effectively to sudden market shocks on the demand side (e.g., the need for food stocks, oil, yeast, medical equipment and medicines, chemicals).

The research highlights differences in problems for IoT implementation among FMCG supply chain participants. These models of multiple linear regression describe the intensity of setbacks, especially for the production (2), physical distribution (3), wholesale (4), and retail sectors (5):

$$y = 0.81 + 0.573x_1 + 0.490x_3 + 0.223x_4 + 0.747x_6 + 0.711x_7 \tag{2}$$

$$y = 0.74 + 0.929x_2 + 0.627x_3 + 0.703x_5 + 0.559x_7 \tag{3}$$

$$y = 0.62 + 0.662x_1 + 0.914x_3 + 0.228x_4 + 0.605x_7 + 0.527x_8 \tag{4}$$

$$y = 0.74 + 0.795x_1 + 0.769x_3 + 0.095x_4 + 0.704x_5 + 0.622x_7 \tag{5}$$

In the given equations concerning the regression model (1), the variable $x_8$ represents standardization. Setbacks without statistically significant impact were not presented. For clarity, the regression equations were graphically represented in Figure 2.

From the graphic presentation above, it seems that the competent institutions and economic policymakers must apply a set of measures and incentives to reduce the peaks of the tested setbacks and bring their values closer to zero. Without incentives and assistance, FMCG supply chains will not be sustainable and will not be able to meet the needs of the market during, as well as after, the COVID-19 pandemic. FMCG WB supply chains were not prepared for the COVID-19 pandemic and the disturbances it caused in the market. In order to reorient the supply chain management towards the digitalization process and to make the supply of FMCG competitive in the crisis, the competent institutions, professional associations, economic policymakers, and economic organizations in the WB region should take the following three groups of measures:

1.  Financial incentives—Imply taking a whole set of measures to financially empower the supply chain participants for the implementation of IoT. These would be specialized credit lines for the

introduction of advanced technology (e.g., low-interest rates, long repayment period), tax relief and exemptions from VAT on devices and equipment for the implementation and application of IoT, deferral of payment for primary energy sources (electricity, water, gas), programs for co-financing the development of modern information infrastructure (e.g., co-financing by National Funds, commercial banks).

2. Incentives for employment—Are related to measures aimed at empowering human resources and training them for the application of advanced technology. These measures imply subsidies for new employment for participants who are accessing digitalization. They also include reduction of taxes and contributions for existing workers, organization of specialized courses and training programs for the use of advanced technologies, education of employees on safety, hygiene, and health measures during the COVID-19 pandemic, knowledge transfer, etc.

3. Security measures—Are all those measures that guarantee the security and privacy protection of users. They include clearly defined privacy policy rules for all IoT users, the introduction of security measures when transferring data (e.g., application of SSL—Secure Socket Layer encryption), strictly defined rules when passing data to third parties, the introduction of security measures to protect networking processes within a chain from malware and external attacks (e.g., installation of firewalls, high-security codes), the assistance of experts and institutions in process standardization, etc. In addition to these measures, state institutions, as well as the participants in the supply chain in the WB region, should raise awareness among producers, distributors, wholesalers, and retailers, as well as consumers, about the need for digitalization and its advantages over traditional supply chains, especially during severe market disruptions and long-term shocks caused by the COVID-19 pandemic.

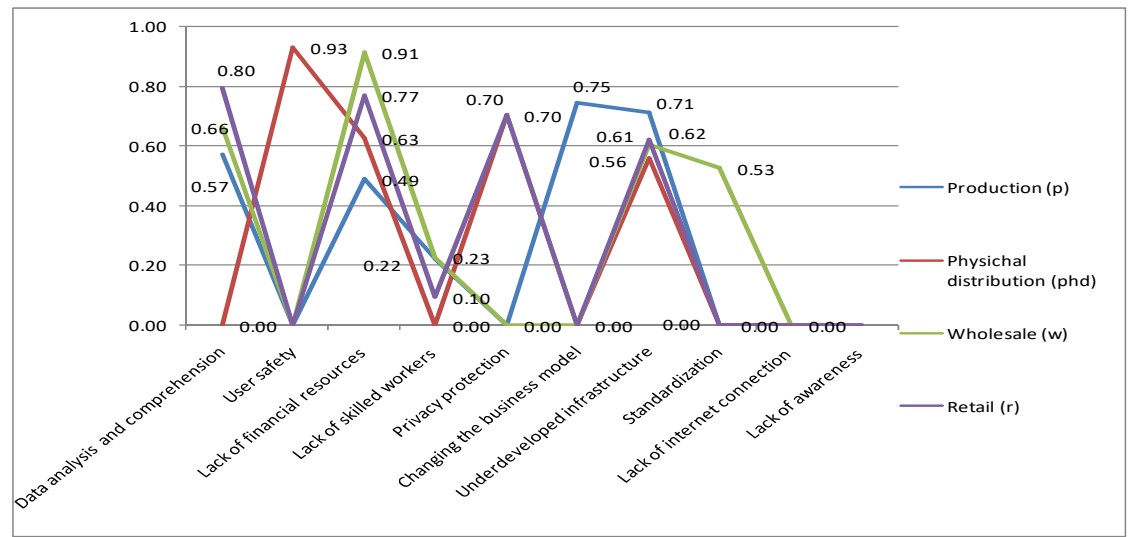

**Figure 2.** Comparison of setbacks to IoT implementation in FMCG supply chains.

*5.2. Comments on the Results*

The obtained results confirmed the findings of previous research conducted on the global market on setbacks and problems faced by supply chain participants in the process of IoT implementation [17,36–39,42,43]. The biggest problem of digitalization of supply chains in high investments, lack of trained workers for successful implementation and maintenance of IoT, and lack of modern infrastructure to support IoT in terms of modern IT equipment, smart devices, artificial intelligence, etc. The obtained results reflect the specificity of the WB market, where the lack of financial resources, insufficient investment in IT infrastructure, and the lack of mentors and experts capable of transferring knowledge, directly limits the modernization of the FMCG supply chain.

Compared to developed countries, this requires the adoption of a comprehensive FMCG supply chain digitalization strategy based on state subsidies and assistance, foreign investment, knowledge transfer from developed supply systems, and a change in the supply chain management philosophy.

The obtained results also confirmed the results of some previous studies on the existence of significant differences between supply chain participants in terms of acceptance of advanced technologies [12,14,17–19,32,35,49,50]. This finding suggests that measures and incentives must be targeted individually at supply chain participants, precisely defining critical points in their business activities and objective reasons for delaying the implementation of IoT in each sector. Only then should the entire supply chain be approached in terms of taking measures to achieve its greater flexibility, transparency, and sustainability.

Surprisingly, the results have not been confirmed by countries. All setbacks are equally present throughout the WB region regardless of economic developments in domestic markets. It suggests that FMCG supply chains function as a single entity throughout the WB region, standardizing placement mechanisms regardless of market characteristics and consumer habits. Therefore, it is necessary to harmonize legal acts, regulations, measures, and incentives related to the process of digitalization of the supply chain among the WB countries, which would provide a stable starting point for the implementation of IoT.

The shortcoming of research is reflected in the geographical limitation of the sample of respondents to the WB region. Objective reasons for choosing such a research sample are the authors' familiarity with the way of placing products and services in the WB region, good institutional cooperation with regional FMCG supply chains, and the availability of data for analysis. The disadvantage may be the questionnaire itself, which consisted of the offered indicators with answers prepared in advance, which could lead to simplified conclusions.

## 6. Conclusions

The COVID-19 pandemic pointed to all the problems which FMCG supply chains face in the WB region, which reflected themselves in the inflexibility and inability to meet the sudden market demand for basic foodstuffs on time. On the other hand, the lockdown policy has significantly jeopardized the efficiency of the functioning of the chain due to the reduction of staff capacity (illness, self-isolation, etc.), the impossibility of physical contacts, rigorous security measures and procedures. The COVID-19 pandemic pointed to an even greater need for a supply system that is less dependent on the human factor, and at the same time totally transparent and efficient. Such a system implies the implementation of digitalization in all production and service operations and supply chain activities, using advanced technologies and devices based on IoT. Since WB countries are at the start of IoT implementation, a need for research on the reasons for such a state of FMCG supply chains arose.

The conducted research defined the importance of various setbacks that delay the implementation of IoT. Setbacks are of varying intensity among supply chain participants, but not between the observed WB countries, which can be treated as a single entity. Based on the defined importance of setbacks, a whole range of measures, incentives, and subsidies have been proposed that should be activated in the WB region to encourage the IoT-based digitalization process. At the time of writing and conducting the research, it is difficult to predict with certainty the course of the COVID-19 pandemic and the economic impact of the crisis that will occur after its end. That is the reason why the management of the FMCG chain must completely change its business philosophy, digitalize business models, and build an intelligent placement system minimally dependent on the human factor.

As part of the proposal for future research, and in line with the indicated shortcomings, the existing research should be extended to countries outside the WB. In this context, a comparison of EU/non-EU countries would be desirable. Furthermore, the structure of the sample of respondents should include top managers, administrative workers, as well as frontline workers (e.g., sales personnel, drivers, warehouse managers). When it comes to the questionnaire, the question structure should include more indicators and specific subcategories. Such an approach to research would complement the scientific

view of the problems and setbacks that arise during the implementation of IoT, and in that context, it would propose a more efficient set of measures to minimize them.

**Author Contributions:** Conceptualization, J.K. and A.G.; methodology, R.M.; investigation, G.V. and S.V.; resources, A.G.; data curation, G.V.; writing—original draft preparation, R.M. and S.V.; visualization, R.M.; supervision, J.K. and A.G. All authors have read and agreed to the published version of the manuscript.

**Funding:** This research received no external funding.

**Conflicts of Interest:** The authors declare no conflict of interest. The funders had no role in the design of the study; in the collection, analyses, or interpretation of data; in the writing of the manuscript, or in the decision to publish the results.

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
