# Peer review of "Setbacks to IoT Implementation in the Function of FMCG Supply Chain Sustainability during COVID-19 Pandemic"

_sustainability, doi:10.3390/su12187391_

Round 1

Reviewer 1 Report

The paper seems to contain interesting material. However, several parts must be improved. See, for instance, my comments below.

- Please, create/add a new subsection 2.1 explain and enumerate the goals and the contributions of the work.

- Please, improve the explanation in the caption of Fig. 1.

- Please, clarify which regression models you have used in Section 5.

- Please, improve the state-of-the-art discussion in the introduction, talking about important and relevant regressor and classification models such as  Gaussian processes and other schemes (such as advanced neural networks), e.g.,

J. Read, et al, "Efficient Monte Carlo Methods for Multi-Dimensional Learning with Classifier Chains", Pattern Recognition, Volume 47, Issue 3, Pages: 1535-1546, 2014.

G. Camps-Valls, et al., "Physics-Aware Gaussian Processes in Remote Sensing", Applied Soft Computing, Volume 28, Pages: 69-82, 2018.

This discussion can improve the number of interested readers and the impact of your paper.

- Add a subsection "Comments to results" in Section 5, giving specific comments, conclusions and observations.

Reviewer 2 Report

This paper proposes a thorough review and analysis on how the Internet of Things (IoT) technology would help in mitigating issues faced by fast-moving-consumer-goods (FMCG) supply chains. The paper is well written and very easy to read. 

I would suggest accepting the paper after some minor revisions:

  • Please cite tables and figures, etc. directly, i.e. avoiding using ‘following’, etc. to refer to a table or a figure.
  • I would suggest moving the discussion of the ‘shortcomings’ of the paper into the discussion section.
